# Transcriptome profiling uncovers differentially expressed genes linked to nutritional quality in vegetable soybean

Xueyang Wang[1ʘ], Chunlei Zhang[1ʘ], Rongqiang Yuan[1ʘ], Xiulin Liu[1], Fengyi Zhang[1], Kezhen Zhao[1], Min Zhang[1], Ahmed M. Abdelghany[2], Sobhi F. Lamlom[3], Bixian Zhang[4]*, Qiang Qiu[5]*, Jia Liu[5]*, Wencheng Lu[6]*, Honglei Ren[1]*

1 Soybean Research Institute of Heilongjiang Academy of Agriculture Sciences, Harbin, China, 2 Crop Science Department, Faculty of Agriculture, Damanhour University, Damanhour, Egypt, 3 Plant Production Department, Faculty of Agriculture Saba Basha, Alexandria University, Alexandria, Egypt, 4 Institute of Biotechnology of Heilongjiang Academy of Agricultural Sciences, Harbin, China, 5 Soybean Research Institute of Jilin Academy of Agriculture Sciences (Northeast Agricultural Research Center of China), Changchun, China, 6 Heihe Branch Institute of Heilongjiang Academy of Agricultural Sciences, Heihe, China

ʘ These authors contributed equally to this work.
* hljsnkyzbx@163.com (BZ); qiuqiang051179@yeah.net (QQ); Lj_hope0459@aliyun.com (JL); 13845674288@163.com (WL); renhonglei2022@163.com (HR)

**Data Availability Statement:** All relevant data are within the manuscript and its Supporting Information files.

## Abstract

Vegetative soybean (maodou or edamame) serves as a nutrient-rich food source with significant potential for mitigating global nutritional deficiencies. This study undertook a thorough examination of the nutritional profiles and transcriptomic landscapes of six soybean cultivars, including three common cultivars (Heinong551, Heinong562, and Heinong63) and three fresh maodou cultivars (Heinong527, HeinongXS4, and HeinongXS5). Nutrient analysis of the seeds disclosed notable differences in the levels of protein, fat, soluble sugars, vitamin E, calcium, potassium, magnesium, manganese, iron, and zinc across the cultivars. Through comparative transcriptome profiling and RNA sequencing, distinct variations in differentially expressed genes (DEGs) were identified between fresh and traditional maodou cultivars. Functional enrichment analyses underscored the involvement of DEGs in critical biological processes, such as nutrient biosynthesis, seed development, and stress responses. Additionally, association studies demonstrated robust correlations between specific DEG expression patterns and seed nutrient compositions across the different cultivars. Sankey diagrams illustrated that DEGs are strongly linked with seed quality traits, revealing potential molecular determinants that govern variations in nutritional content. The identified DEGs and their relationships with nutritional profiles offer valuable insights for breeding programs focused on developing cultivars with improved nutritional quality, tailored to specific dietary needs or industrial applications.

**Funding:** funded by Scientific Research Business Expenses of Heilongjiang Scientific Research Institutes (Grant No. CZKYF2021-2-C030; Grant No. CZKYF2024-1-A003; Grant No. CZKYF2022-1-A002; Grant No. CZKYF2023-1-A001); Innovation Leaping Project in Heilongjiang Academy of Agricultural Science (Grant No. CX23ZD04; Grant No. CX23TS04; Heilongjiang Natural Science Foundation joint guidance project (LH2020C093). The funders had no role in study design, data collection and analysis, decision to publish, or preparation of the manuscript.

**Competing interests:** The authors have declared that no competing interests exist.

## 1. Introduction

For more than 2000 years, soybean (*Glycine max*) has been an integral part of East Asian agriculture and cuisine [1]. Soybean seeds are rich in protein, oil, and carbohydrates, with up to 47% of the carbohydrate content consisting of soluble sugars [2]. Soybean protein is among the most cost-effective sources of dietary protein and is regarded as an acceptable substitute for animal-derived proteins because of its complete essential amino acid profile, including lysine, which is deficient in many cereal staples [3–5]. Moreover, soybean oil has a widespread preference because of its high polyunsaturated and low saturated fatty acid composition [5,6]. In addition to protein and oil, soybeans also constitute a rich source of a wide spectrum of vital nutrients, bioactive components, phytonutrients, and minerals [7,8]. Consequently, the soybean crop has maintained an integral role in Asian culinary traditions for millennia [9].

The soybean crop exhibits versatility, enabling its transformation into diverse food products and facilitating consumption at various maturation stages, encompassing the sprouting, vegetable, and mature phases [10]. The widely known mature soybean is harvested in a dry state and processed into an array of soy-derived foods, such as tofu, soy sauce, soybean milk, natto, and numerous others [11]. Conversely, vegetable soybeans, referred to as "Maodou" in China and "Edamame" in Japan, are harvested at the immature stage (R6–R7) when the seed occupies 80–90% of the pod cavity [12,13]. While mature soybean seeds primarily undergo processing before consumption, potentially resulting in nutrient losses, vegetable soybean is consumed as a whole seed, either as a snack or a vegetable [14]. As a snack, maodou pods undergo brief cooking in salted water for several minutes, whereas when consumed as a vegetable, the seeds can be fried with meat or other vegetables or subjected to a short boiling period for incorporation into salads [15]. Consequently, the reduced cooking times and whole-seed consumption associated with maodou may significantly limit nutritional losses, underscoring its potential importance for dietary incorporation [16,17]. In recent years, maodou consumption has gained popularity in Western countries, particularly in the USA, owing to its high nutritional value [17,18]. For this purpose, the nutritional composition of maodou, including folates, carotenoids, tocopherols, and minerals, was recently analyzed [13,19,20]. Moreover, the nutritional impact of consuming 100 g of maodou per day was assessed, alongside an analysis of the expression profiles of key biosynthetic enzymes responsible for regulating critical soybean quality traits during seed development.

Interestingly, metabolomic and transcriptomic analyses have been successfully employed to study soybean seed development and composition [21–24]. For example, Lin et al. [21] detected 169 metabolites across 29 soybean cultivars, with 104 exhibiting differential abundances. Chebrolu et al. [25] analyzed 275 seed metabolites under heat stress and reported enrichment of antioxidant metabolites in heat-tolerant genotypes. Sun et al. [26] utilized transcriptome data from eight seed tissues across three developmental stages, demonstrating that tissue-specific gene expression was more prominent than stage-specific expression, and they identified key regulatory networks along with hub genes. Previous RNA-seq studies on four chromosome substitution lines with differing oil and protein contents across three seed developmental stages were conducted [27,28]. Through weighted gene co-expression network analysis, these studies identified 46 expression patterns, and seven hub genes associated with oil and protein accumulation. Furthermore, transcriptome studies in contrasting seed size lines have implicated hormonal signaling pathways and transcription factors as key regulators of seed size determination [29,30]. By conducting a comparative transcriptome analysis between cultivated and wild soybean lines, researchers identified 2,680 differentially expressed genes (DEGs) [31]. Integrating transcriptomic data with quantitative trait locus (QTL) information, they further delineated two cultivar-specific gene co-expression networks. Furthermore, this

integrative approach facilitated the identification of two key genes, GA20OX and NFYA, which were found to play crucial roles in the regulation of seed trait development and formation [32,33]. These findings highlight the utility of combining transcriptomic profiling with genetic mapping techniques to uncover the molecular underpinnings governing seed characteristics in soybean [34]. For instance, a combined metabolomic and transcriptomic analysis performed across five developmental stages of soybean seeds uncovered 148 metabolites and 2,869 differentially expressed genes (DEGs) linked to variations in seed traits, revealing associated genes, networks, and metabolic alterations [35].

However, a substantial gap remains in understanding the differences in metabolite abundance and regulatory networks between grain and vegetable soybean seed development. Comprehensive studies are needed to clarify the metabolic and transcriptional differences driving these developmental processes. This study provides a novel and in-depth examination of the nutritional profiles of maodou soybean cultivars. We analyzed six distinct cultivars: Heinong551, Heinong562, and Heinong63, representing common soybeans, as well as Heinong527, HeinongXS4, and HeinongXS5, which are classified as fresh maodou varieties. By performing a comparative transcriptomic analysis of fresh seeds at the R6 developmental stage, we aimed to identify variations in seed composition and uncover the underlying gene regulatory networks pertinent to vegetable soybean. Our findings contribute to an enhanced understanding of the molecular mechanisms governing seed development and provide valuable insights to guide future molecular breeding efforts focused on the genetic improvement of vegetable soybean cultivars.

## 2. Materials and methods

### 2.1 Plant materials and experimental design

Six soybean (*Glycine max* L. Merr.) cultivars were utilized in this study: Heinong551, Heinong562, and Heinong63, representing common varieties, and Heinong527, HeinongXS4, and HeinongXS5, classified as fresh maodou varieties. All cultivars were obtained from the Heilongjiang Academy of Agricultural Sciences, Heilongjiang Province, China. The field experiment was conducted in 2023 at the experimental station of the Heilongjiang Academy of Agricultural Sciences. Sowing was performed on May 10th, 2023, using a randomized complete block design with three replications. Each cultivar was planted in four-row plots, with rows measuring 8 meters in length and spaced 0.65 meters apart. The planting density was standardized at 225,000 plants per hectare (equivalent to 15,000 plants per mu in the Chinese land area system). Fresh grain samples were harvested on September 3rd, 2023, corresponding to the R6 developmental stage (full seed stage) of soybean growth. This stage is characterized by pod cavities filled with seeds that have reached their full size.

### 2.2 Determination of protein, fatty acids, soluble sugars, vitamin E, and mineral content

Crude protein content was assessed using the Kjeldahl method to measure total nitrogen, which was then converted to protein content by applying a conversion factor of 6.25 [36]. Fatty acids were extracted by placing 30 mg of soybean powder in screw cap tubes with 1 ml of 2.50% $H_2SO_4$. The samples were heated at 85°C for 1 hour and 30 minutes, with shaking every 10 minutes, followed by a 10-minute cooling period. Subsequently, 150 μL of NaCl and 700 μL of hexane were added, and the samples were shaken for 3 minutes. They were then centrifuged at 4000 rpm for 10 minutes, and the supernatant was collected in 2 ml vials for analysis by gas chromatography [37]. Soluble sugars were quantified according to the method of Jie et al. [38].

In brief, 100 mg of soybean powder was mixed with 50% acetonitrile and shaken for 8 hours at room temperature in an incubator. A 500 μL aliquot of the supernatant was transferred to a new tube containing 200 μL of acetonitrile, shaken to precipitate proteins, and left at room temperature for 10 minutes. The samples were then centrifuged at 20°C for 10 minutes, and the supernatant was filtered through 0.22 μm syringe filters before analysis by UPLC-RID. Vitamin E extraction and quantification followed the procedure outlined by Ghosh et al. [39].

Minerals were extracted by weighing 0.10 g of dried soybean powder and mixing it with 4 ml of concentrated nitric acid. The mixture was digested using a super microwave digestion system (Hangzhou Puyu Company). Following dilution, the samples were analyzed using ICP-MS (Hangzhou Puyu Company) to quantify the levels of magnesium (Mg), potassium (K), calcium (Ca), manganese (Mn), iron (Fe), and zinc (Zn).

## 2.3 RNA preparation and RNA sequencing

Total RNA from six soybean cultivars, namely, Heinong551 (HN551), Heinong562 (HN562), Heinong63 (HN63), Heinong527 (HN527), HeinongXS4 (XS4hao) and HeinongXS5 (XS5hao), was extracted via TRIzol (Invitrogen, CA, USA). Each cultivar had three biological replicates, resulting in 18 total samples. The quantity of RNA was verified via 1.0% agarose gel electrophoresis, and the concentration was measured via a 2100 Bioanalyzer (Agilent Technologies, Santa Clara, CA, USA). The extracted total RNA with an RNA integrity number >7.0 was used to construct cDNA libraries with an NEBNext UltraTM RNA Library Prep Kit for Illumina (NEB, Beijing, China). Libraries were sequenced on an Illumina NovaSeq 6000 Sequencer (Illumina Inc., California, USA). The clean reads were mapped to the reference genome sequence of soybean (https://www.ncbi.nlm.nih.gov/datasets/genome/GCF_000004515.6/) via STAR (v2.7.10a) software. The raw reads were submitted to the sequence read archive of the National Center for Biotechnology Information (NCBI) under BioProject ID No: PRJNA1128828 (https://www.ncbi.nlm.nih.gov/bioproject/?term=PRJNA1128828). The following databases were used for functional annotation: Gene Ontology (GO) (http://geneontology.org/) and KEGG (Kyoto Encyclopedia of Genes and Genomes, http://www.genome.jp/kegg/).

## 2.4 Differential expression analysis and functional enrichment

To identify differentially expressed genes (DEGs) among samples, transcript expression levels were calculated using the fragments per kilobase of exon per million mapped reads (FPKM) method. Gene transcript abundances were quantified with RNA-Seq by Expectation-Maximization (RSEM v1.3.2) (http://deweylab.biostat.wisc.edu/rsem/). Differential expression analysis was conducted in R (version R-4.0.5) using the package edgeR for empirical analysis of digital gene expression. DEGs were determined through edgeR software, with p-values adjusted using the False Discovery Rate (FDR) method. Genes with an FDR < 0.05 and p-value < 0.05 were classified as DEGs. Pathway enrichment analysis was performed with clusterProfiler (v4.8.2) in R (version R-4.0.5), adopting Benjamini-Hochberg (BH) adjustment method, considering only genes with p.adjust < 0.05 as significantly enriched.

The Mantel test was conducted using the "vegan" package (version 2.5–7) in R (version R-4.0.5). Statistical significance was determined with the following thresholds: *, p < 0.05; **, p < 0.01; ***, p < 0.001. Correlation and co-occurrence network analyses were performed using the Spearman method with BH adjustment for p-values. Line thickness in the co-occurrence network represents the magnitude of correlations, and dot size reflects relative abundance.

Sankey diagrams were generated to illustrate DEGs and their associations with seed quality traits in three fresh maodou soybean cultivars compared with the common group. The diagrams were constructed using the "networkD3" package in R (version R-4.0.5). The analysis

included five columns: individual DEGs, clusters of co-expressed DEGs, correlations with Spearman's p value < 0.05 and |rho| > 0.7, and specific seed quality parameters showing significant differences relative to the common cultivars. The Sankey diagrams visually represent the relationships between DEGs, their expression patterns within clusters, and their contributions to variations in seed nutrient composition, including protein, fat, soluble sugar, vitamin, and mineral levels.

## 2.5 Relative expression analysis

Quantitative real-time PCR (qRT–PCR) validation was performed on thirty-four randomly selected genes from RNA sequencing data. Seed samples from the six soybean cultivars were collected in triplicate. Total RNA was extracted from these samples using an RNA isolation reagent (Thermo Fisher Scientific, USA). First-strand cDNA was synthesized with TransScript® One-Step gDNA Removal and cDNA Synthesis SuperMix (Transgen, China). Gene-specific primers for qPCR were designed using the Primer3 website (**S1 Table**), with the soybean GmActin gene (Glyma.18G290800) serving as the internal reference gene (S1 Table). The qRT–PCR reactions were conducted using the CFX96 Touch Real-Time PCR Detection System (Bio-Rad, USA) with ChamQ SYBR qPCR Master Mix (Vazyme, China). For gene expression and transcriptome analyses, three independent samples were collected from different points within each block for each cultivar. These three samples were pooled to create one representative biological replicate, resulting in three biological replicates per cultivar (one pooled sample per block). This composite sampling strategy, covering nine total sampling points per cultivar across the field (three sampling points × three blocks), was implemented to account for field variation while maintaining statistical power. A total of 18 pooled samples (6 cultivars × 3 biological replicates) were processed for RNA sequencing analysis.

Hierarchical clustering and heatmap visualization of gene expression profiles were conducted for vegetable soybean cultivars, focusing on 34 genes. Gene expression data from these 34 genes across four cultivars (HN551, HN562, HN63, and XS4hao) were analyzed using the "pheatmap" R package. For hierarchical clustering, the Euclidean distance metric was used to calculate dissimilarities between gene expression profiles. Clustering was conducted using the Ward.D2 method, which optimizes the variance within clusters. The heatmap visualizes expression patterns with a color gradient, where blue indicates low expression and red indicates high expression. The clustering results are illustrated by a dendrogram, which displays the hierarchical relationships among genes and cultivars.

# 3. Results

## 3.1 Comparative nutritional profiles of common varieties and fresh maodou soybean cultivars

To compare the nutritional profiles of the six cultivars—common soybeans (Heinong551, Heinong562, and Heinong63) and fresh maodou soybeans (Heinong527, HeinongXS4, and HeinongXS5) (**Fig 1**)—we analyzed their seed nutrient compositions. This analysis included measurements of protein, fat, soluble sugar, vitamin E, and the mineral contents of Ca, K, Mg, Mn, Fe, and Zn, as shown in **Fig 2**. Our analysis revealed significant differences across all the examined parameters of seed nutritive composition among the six soybean cultivars investigated (**Fig 2**). A significant difference in protein content was observed among the cultivars (p value < 0.001). HN527 presented the highest protein content (14.37%), which was significantly greater than that of HN551 (10.23%), the cultivar with the lowest protein content. For fat content, highly significant differences were also detected in fat content among the cultivars (p

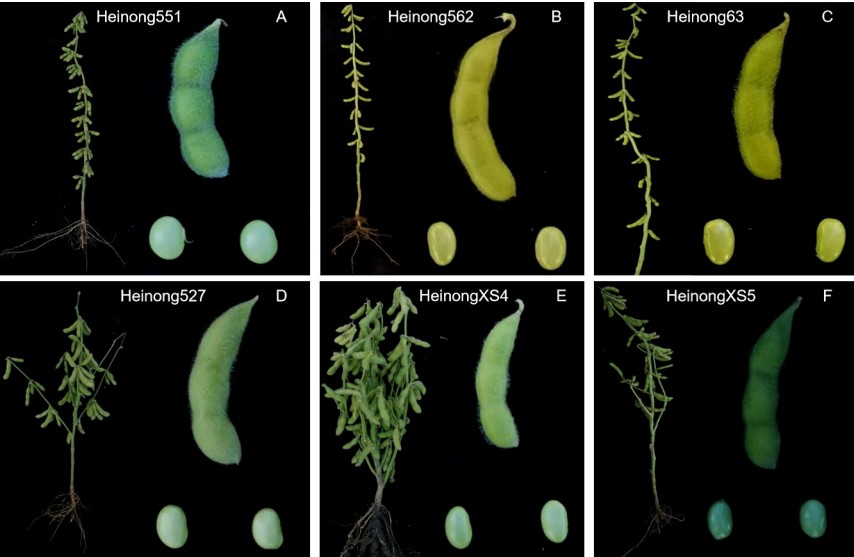

**Fig 1. Morphological comparison of six soybean cultivars used in the study.** Panels A to F display the whole plant, pod, and seed of each cultivar. The common soybeans are represented by Heinong551 (A), Heinong562 (B), and Heinong63 (C), while the fresh maodou soybeans are represented by Heinong527 (D), HeinongXS4 (E), and HeinongXS5 (F).

value < 0.001). The highest fat content was obtained from the soybean cultivar HN527 (9.71%), whereas that of HN63 was only 2.19%, indicating the lowest fat content.

The HN527 cultivar presented the highest significant levels of soluble sugar, Ca, and K among the six soybean cultivars investigated. Specifically, HN527 recorded a soluble sugar content of 9.71%, a Ca concentration of 823.33 mg/kg, and a K level of 9569 mg/kg. On the other hand, the HN551 cultivar presented the lowest soluble sugar content of 2.09% and a Ca concentration of 445.67 mg/kg, whereas the HN562 cultivar presented the lowest K level of 7161.67 mg/kg. Furthermore, the nutritional analysis revealed varying levels of vitamin E, magnesium and iron among the soybean cultivars tested. XS4hao contained the highest amounts of vitamin E at 1.09 mg/g, magnesium at 1111.67 mg/kg, and iron at 97.37 mg/kg. In contrast, HN562 had the lowest magnesium content (778.33 mg/kg) and iron level (30.67 mg/kg). Moreover, compared with the other cultivars, HN527 presented the lowest vitamin E concentration of 0.74 mg/g. For the Mg and Zn levels, XS5hao contained the highest amounts of Mn (42.53 mg/kg) and Zn (17.5 mg/kg). In contrast, HN527 presented the lowest levels of both manganese and Zn, with Mn quantified at 8.09 mg/kg and zinc at 12.47 mg/kg, compared with those of the other cultivars tested. The results demonstrate considerable variation in seed nutrient composition among the six soybean cultivars studied. These findings have important implications for breeding programs aimed at developing cultivars with enhanced nutritional profiles tailored to specific dietary needs or industrial applications.

### 3.2 RNA-seq data and mapping for transcriptome profiling of soybean cultivars

In the present study, we investigated the transcriptomic profiles of six soybean cultivars, namely, Heinong551 (HN551, A), Heinong562 (HN562, B), Heinong63 (HN63, C), Heinong527 (HN527, D), HeinongXS4 (XS4hao, E), and HeinongXS5 (XS5hao, F). To elucidate the molecular mechanisms underlying the phenotypic variations between these two soybean groups, we employed RNA-Seq technology to analyze gene expression patterns.

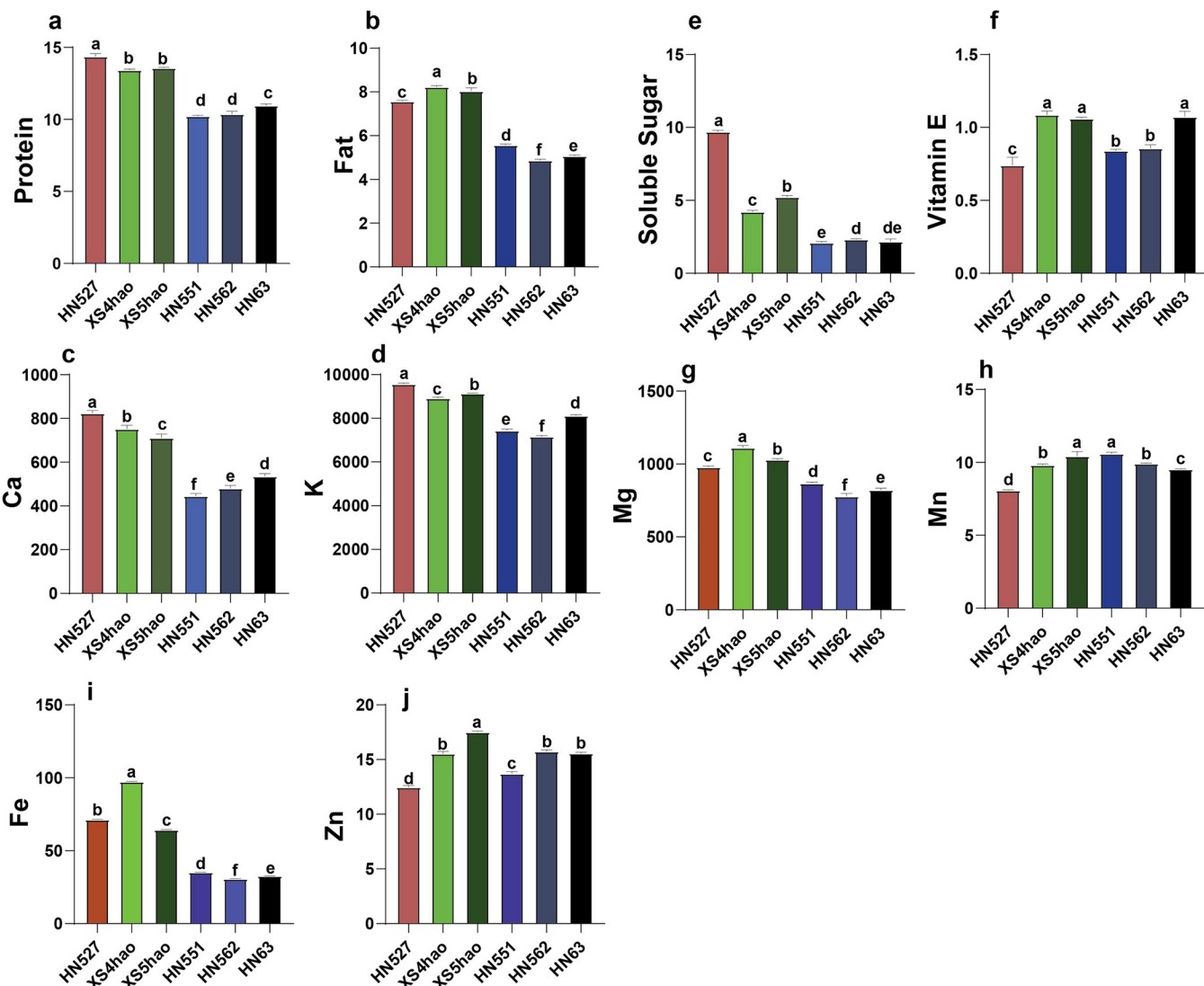

**Fig 2. Seed nutrient contents of the six soybean cultivars.** The values are the means ± standard errors (n = 3). Different lowercase letters above the bars indicate significant differences among cultivars for a given nutrient on the basis of analysis of variance (ANOVA) and post hoc tests (p < 0.001).

For each cultivar, three biological replicates were utilized, resulting in a total of 18 samples. Illumina sequencing of the 18 RNA libraries generated an average of 48.5 million 150-bp paired-end reads per sample, ranging from 39.5 million to 73.8 million reads (**S2 Table**). Quality control measures, including filtration, were applied to obtain approximately 867.7 million high-quality cleaned reads, which presented a GC content within the acceptable range of 44.76% to 46.36% (**S3 Table**). Subsequent alignment of the filtered reads to the soybean reference genome revealed an average mapping rate of 94.27% for reads with Phred quality scores ≥ 30, spanning 82.33% to 92.60% across samples (**S4 Table**).

## 3.3 Comparative transcriptome analysis reveals DEGs distinguishing common and fresh maodou soybean cultivars

To identify DEGs between common soybean cultivars (A, B, and C) and fresh maodou cultivars (D, E, and F), 9 pairwise comparisons were performed (i.e., D_vs_A, D_vs_B, D_vs_C,

E_vs_A, E_vs_B, E_vs_C, F_vs_A, F_vs_B, and F_vs_C). The findings of comparative transcriptome analysis revealed that a total of 66,778 genes were mapped to the soybean genome, and the number of DEGs identified in each comparison group was as follows: 176 DEGs (49 upregulated and 127 downregulated) in D_vs_A, 246 DEGs (88 upregulated and 158 downregulated) in D_vs_B, 202 DEGs (113 upregulated and 89 downregulated) in D_vs_C, 915 DEGs (457 upregulated and 458 downregulated) in E_vs_A, 655 DEGs (295 upregulated and 360 downregulated) in E_vs_B, 2,304 DEGs (1,119 upregulated and 1,185 downregulated) in E_vs_C, 822 DEGs (584 upregulated and 238 downregulated) in F_vs_A, 282 DEGs (155 upregulated and 127 downregulated) in F_vs_B, and 1,145 DEGs (766 upregulated and 379 downregulated) in F_vs_C (S4 Table).

A deeper exploration of the key DEGs across different comparison groups revealed variations in the magnitude of gene expression differences. By applying a stringent filtering threshold of $|log2FC| \geq 3$ and p value $< 0.01$, 29 key genes located at the extreme ends of the volcano plot were obtained when the D cultivar was compared with the common cultivars (A, B, and C) (Fig 3A). Similarly, 34 key genes and 31 key genes were identified at the top and bottom of the volcano plots for the E cultivar (Fig 3C) and F cultivar (Fig 3E), respectively, compared with those of common cultivars.

Additionally, to identify the core set of DEGs, Venn diagrams were constructed to visualize the overlap of upregulated and downregulated DEGs among the three comparison groups. Additionally, heatmaps were generated to illustrate the expression patterns of those overlapping DEGs. When the D cultivar was compared with the common cultivar, 17 DEGs were upregulated, and 32 DEGs were downregulated (Fig 3B). For the E cultivar, 60 DEGs were upregulated, and 106 DEGs were downregulated compared with those in the common (Fig 3D). In the F cultivar, 88 DEGs were upregulated, and 66 DEGs were downregulated compared with those in the common (Fig 3F). Overall, our analysis identified 39 DEGs, including 29 key genes, that exhibited distinct upregulated and downregulated expression trends in the D cultivar compared with the common cultivar. Additionally, 166 DEGs, encompassing the 34 key genes, displayed an obvious upregulated and downregulated pattern in the E cultivar, whereas 154 DEGs, including the 31 key genes, presented a similar trend in the F cultivar compared with the common cultivars.

## 3.4 Functional enrichment of the DEGs in different cultivars via GO and KEGG analyses

GO enrichment analysis of upregulated and downregulated DEGs was conducted via the GO database. The DEGs of the three fresh maodou enriched by the filtering threshold p. adjust$<0.05$ and p value$<0.01$ (Fig 4A). In total, 39 DEGs were significantly enriched in 23 GO pathways when comparing D with A, B, and C. Among these, seven upregulated DEGs were significantly enriched in several key pathways: GO:0004372 (glycine hydroxymethyltransferase activity), which is involved in one-carbon metabolism crucial for glycine and serine synthesis, thereby impacting amino acid and protein production; GO:0016229 (steroid dehydrogenase activity); GO:0004033 (aldo-keto reductase (NADP) activity); and GO:0016742 (hydroxymethyl, formyl and related transferase activity). These enriched pathways suggest possible alterations in carbohydrate metabolism, which may be associated with sugar metabolism and stress reactions, potentially affecting seed carbohydrate concentration and modifying the overall caloric value and sweetness of the soybeans. Additional enriched pathways included GO:0004725 (protein tyrosine phosphatase activity), which regulates protein dephosphorylation and controls signal transduction pathways affecting growth, metabolism, and stress responses; GO:0070300 (phosphatidic acid binding), which involves membrane lipid

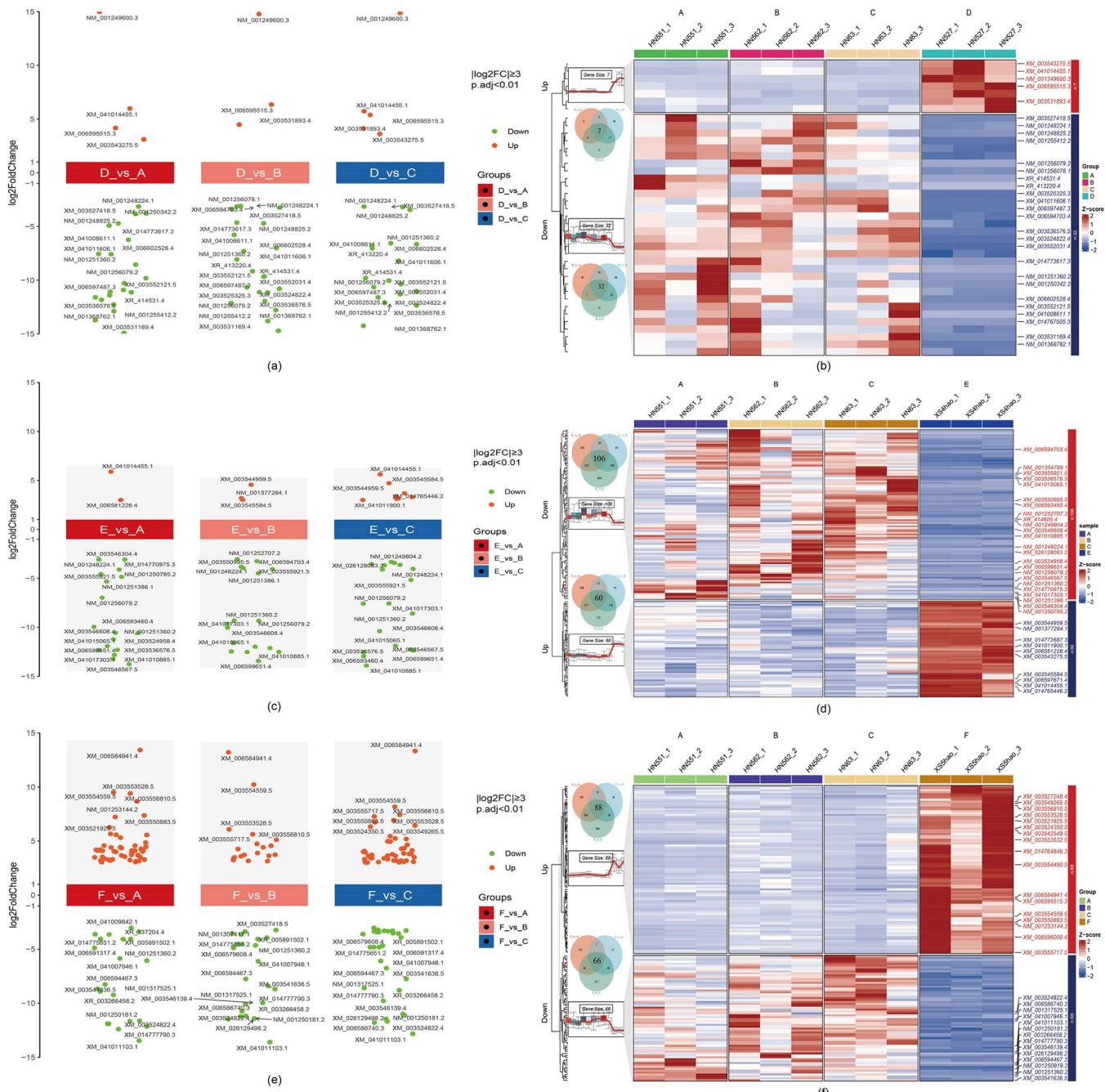

**Fig 3. Identification of key DEGs between fresh maodou and soybean cultivars through comparative transcriptome analysis.** (a, c, e) Volcano plots depict upregulated and downregulated DEGs in the fresh maodou cultivars D, E, and F, respectively, compared with the common cultivars (A, B, and C). Significantly upregulated DEGs (log2-fold change > 1 and p value < 0.05) are shown in red, whereas significantly downregulated DEGs (log2-fold change < -1 and p value < 0.05) are shown in green. Highly significant DEGs were identified by applying a stringent filtering threshold of |log2-fold change| ≥ 3 and p value < 0.01, marked with black text labels. (**b, d, f**) Venn diagram depicting the overlap of upregulated (red) and downregulated (green) DEGs in the D cultivar compared with the common cultivars A, B, and C. Heatmap illustrating the expression patterns of the overlapping DEGs across the comparison groups. In the D cultivar, 7 DEGs were upregulated, whereas 32 DEGs presented downregulated expression compared with the common cultivars (A, B and C). Compared with those in common, 60 DEGs in E were upregulated, and 106 DEGs were downregulated (A, B and C). Compared with those in common, 88 DEGs in F were upregulated, and 66 DEGs were downregulated (A, B and C).

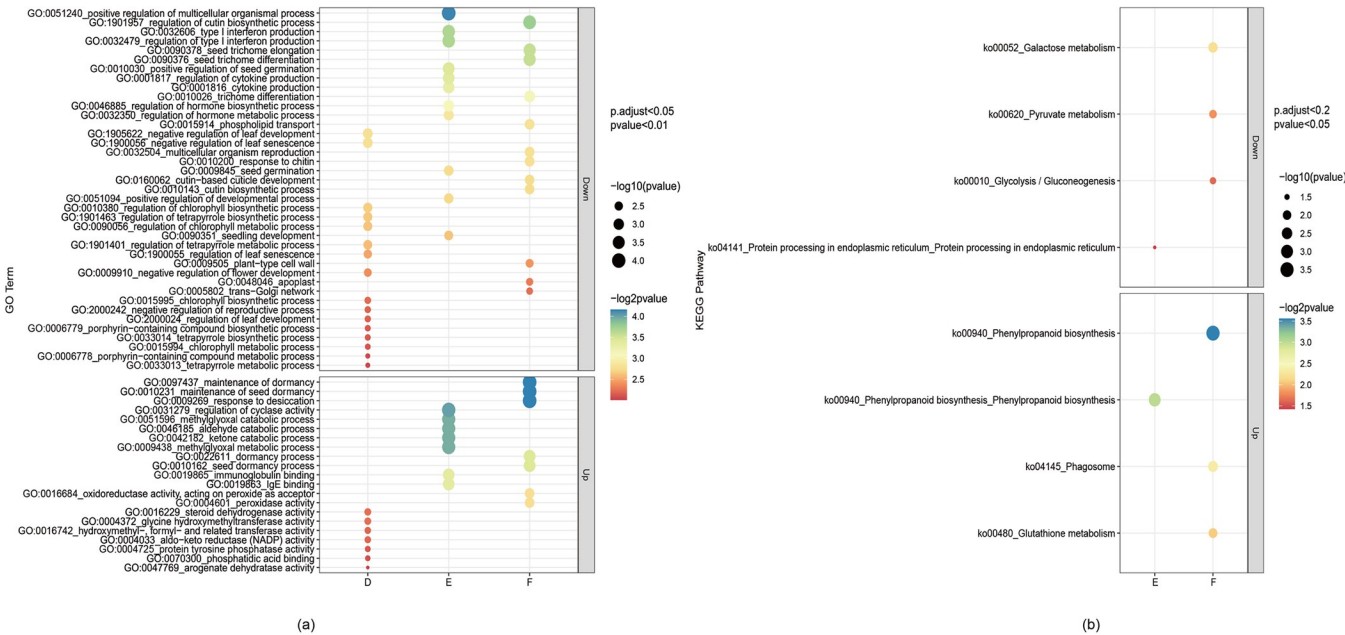

**Fig 4. Functional enrichment analysis of the DEGs.** (**a**) Enrichment of DEGs from three fresh majority samples in the GO database, applying a filtering threshold of p.adjust < 0.05 and p value < 0.01. The dot size represents the level of significance of enrichment. (**b**) Enrichment of DEGs from three fresh maodou samples in the KEGG database, using a filtering threshold of p.adjust < 0.2 and p value < 0.05. The dot size indicates the significance of enrichment.

interactions that potentially affect stress signaling and cellular metabolism; and GO:0047769 (arogenate dehydratase activity), which plays a crucial role in phenylalanine biosynthesis, important for protein synthesis and secondary metabolite production, such as lignin and flavonoids. The remaining 32 DEGs were downregulated and significantly enriched in several GO pathways related to leaf development and photosynthesis: GO:1900056 (negative regulation of leaf senescence), GO:1905622 (negative regulation of leaf development), GO:0010380 (regulation of chlorophyll biosynthetic process), and GO:1901463 (regulation of tetrapyrrole biosynthetic process). These processes maintain leaf functionality and photosynthetic efficiency, supporting sustained growth and nutrient accumulation, which ultimately impacts seed yield and quality. Additional enriched pathways included GO:0090056 (regulation of chlorophyll metabolic process), GO:1901401 (regulation of tetrapyrrole metabolic process), GO:1900055 (regulation of leaf senescence), GO:0009910 (negative regulation of flower development), and GO:0015995 (chlorophyll biosynthetic process). These pathways indicate changes in the control of leaf aging and chlorophyll levels, potentially affecting photosynthetic efficiency. The enhanced photosynthesis throughout the growth period may influence resource allocation, particularly the accumulation of carbohydrates and proteins in the seeds.

When comparing cultivar E with the common cultivars (A, B, and C), we identified 60 upregulated and 106 downregulated DEGs. The upregulated DEGs were significantly enriched in GO:0031279 (regulation of cyclase activity), GO:0009438 (methylglyoxal metabolic process), GO:0042182 (ketone catabolic process), GO:0046185 (aldehyde catabolic process), GO:0051596 (methylglyoxal catabolic process), GO:0019863 (IgE binding), and GO:0019865 (immunoglobulin binding). The methylglyoxal-related processes are involved in detoxifying harmful metabolic byproducts, potentially influencing stress tolerance and nutrient balance. The GO terms related to ketone, aldehyde, and methylglyoxal catabolic processes suggest involvement in breaking down toxic byproducts from metabolism, which could play a role in

cellular detoxification and stress responses. The enrichment of IgE and immunoglobulin binding may indicate interactions with plant defense mechanisms.

The downregulated DEGs were significantly enriched in pathways related to development and immune responses: GO:0051240 (positive regulation of multicellular organismal process), GO:0032479 (regulation of type I interferon production), GO:0032606 (type I interferon production), and GO:0010030 (positive regulation of seed germination). Additional enriched pathways included GO:0001817 (regulation of cytokine production), GO:0001816 (cytokine production), GO:0046885 (regulation of hormone biosynthetic process), and GO:0032350 (regulation of hormone metabolic process). These processes involve signaling proteins that mediate and regulate immunity, inflammation, and hematopoiesis, possibly influencing plant stress responses. The pathways GO:0009845 (seed germination), GO:0051094 (positive regulation of developmental process), and GO:0090351 (seedling development) support growth and differentiation necessary for seedling establishment and plant maturation, affecting early plant growth and final yield. The downregulation of these genes suggests a potential optimization of energy allocation, possibly favoring growth or nutrient accumulation over developmental and immune responses.

In the F cultivar compared with the common cultivars (A, B, and C), 154 DEGs were significantly enriched in 20 GO pathways. Among the upregulated DEGs, 88 genes were enriched in GO terms related to response to desiccation (GO:0009269), maintenance of seed dormancy (GO:0010231, GO:0097437), seed dormancy process (GO:0010162), dormancy process (GO:0022611), peroxidase activity (GO:0004601), and oxidoreductase activity acting on peroxide as an acceptor (GO:0016684). Conversely, the 66 downregulated DEGs were significantly enriched in GO terms associated with the regulation of cutin biosynthetic process (GO:1901957), seed trichome differentiation and elongation (GO:0090376, GO:0090378), trichome differentiation (GO:0010026), phospholipid transport (GO:0015914), multicellular organism reproduction (GO:0032504), response to chitin (GO:0010200), cutin biosynthetic process (GO:0010143), cutin-based cuticle development (GO:0160062), plant-type cell wall (GO:0009505), apoplast (GO:0048046), trans-Golgi network (GO:0005802), and Golgi apparatus subcompartment (GO:0098791).

KEGG enrichment analysis was conducted to investigate the differential gene expression patterns in fresh maodou samples compared with those in common samples (**Fig 4B**). DEGs meeting the filtering criteria of p.adjust $< 0.2$ and p value $< 0.05$ were considered for analysis. Notably, no significantly enriched KEGG pathways were identified in sample D compared with the common group (A, B, and C). In contrast, in sample E, 166 DEGs were significantly enriched in two KEGG pathways compared with those in the common. Specifically, 60 upregulated DEGs were notably enriched in ko00940 (phenylpropanoid biosynthesis), whereas 106 downregulated DEGs were enriched in ko04141 (protein processing in the endoplasmic reticulum). Similarly, in sample F, compared with the common, 154 DEGs were significantly enriched across six KEGG pathways. Among these, 88 upregulated DEGs were enriched in ko00940 (phenylpropanoid biosynthesis), ko04145 (phagosome), and ko00480 (glutathione metabolism), whereas 66 downregulated DEGs were enriched in ko00052 (galactose metabolism), ko00620 (pyruvate metabolism), and ko00010 (glycolysis/gluconeogenesis). The KEGG enrichment analysis revealed significant pathways associated with nutrition production and stress response mechanisms. The upregulated DEGs in phenylpropanoid biosynthesis pathway (ko00940) play crucial roles in facilitating the formation of lignin and flavonoids, which are essential compounds for both plant defense mechanisms and stress resilience. Meanwhile, the downregulation of DEGs involved in fundamental metabolic pathways, including glycolysis/gluconeogenesis (ko00010) and pyruvate metabolism (ko00620), suggests significant alterations in primary metabolism. These changes may influence energy production and

carbohydrate metabolism, potentially affecting both the nutritional quality of the seeds and the plant's capacity to respond to environmental stresses. Understanding these pathway interactions provides valuable insights into how differential gene expression influences both the nutritional composition of vegetable soybean and its adaptation to environmental challenges.

## 3.5 Association of DEGs with nutritional profiles across multiple maodou cultivars

Mantel tests were used to assess the correlations between key DEGs and seed nutrition profile, including protein, fat, soluble sugar, vitamin E, Ca, K, Mg, Mn, Fe, and Zn. The results revealed that the upregulated and downregulated DEGs were significantly correlated with vitamin E in D compared with the common DEGs. Specifically, 32 DEGs that were downregulated in D compared with D were significantly negatively correlated with protein (15/32), fat (12/32), soluble sugar (14/32), Ca (18/32), K (19/32), Mg (13/32) and Fe (15/32) and significantly positively correlated with Mn (16/32), vitamin E (17/32) and Zn (12/32). Additionally, a set of 7 DEGs that were upregulated in D compared with the common group were significantly positively correlated with protein (1/7), fat (4/7), soluble sugar (4/7), Ca (2/7), K (3/7), Mg (4/7) and Fe (4/7) and significantly negatively correlated with Mn (2/7), vitamin E (7/7) and Zn (4/7) (**Fig 5A**).

Compared with the common DEGs, the upregulated DEGs were significantly correlated with vitamins E, K and Mn, and the downregulated DEGs were significantly correlated with protein, fat, vitamin E, Ca, K, Mg and Mn in D. Specifically, 24 DEGs that were downregulated in E compared with the common group presented a significantly negative correlation with vitamin E (6/24), protein (10/24), Ca (10/24), soluble sugar (13/24), K (7/24), Fe (10/24), fat (11/24), Mg (11/24) and Zn (2/24) and a significantly positive correlation with Mn (3/24); the 10 obvious DEGs that were upregulated in E compared with the common group presented a significantly positive correlation with vitamin E (1/10), protein (1/10), Ca (2/10), soluble sugar (5/10), K (3/10), Fe (5/10), fat (6/10) and Mg (5/24); and a significantly negative correlation with Mn (1/10) (**Fig 5B**). The upregulated and downregulated DEGs were significantly correlated with protein, fat, soluble sugar, vitamin E, Ca, K, Mg, Mn, Fe and Zn in F compared with the common group. Additionally, 14 DEGs that were downregulated in F compared with those in common were significantly negatively correlated with vitamin E (5/14), protein (9/14), Ca (10/14), soluble sugar (10/14), K (8/14), Fe (4/14), fat (4/14), Mg (5/14) and Zn (9/14), and the 17 obvious DEGs that were upregulated in F compared with the common genes were significantly positively correlated with vitamin E (1/17), protein (10/17), Ca (11/17), soluble sugar (10/17), K (5/17), Fe (5/17), fat (5/17), Mg (6/17), Mn (4/17) and Zn (11/17) (**Fig 5C**). To investigate the potential reciprocal interactions between key DEGs and quality traits, a co-occurrence network was constructed via Spearman correlation analysis (**Fig 5D**).

After the correlations between the ten seed nutrient parameters and the expression patterns of DEGs across the various soybean cultivars were established, our analysis focused on the genes that exhibited a robust association with the seed quality traits (**Fig 6**). By applying stringent thresholds, considering only those genes with an absolute correlation coefficient (|r|) greater than 0.7 and a p value less than 0.05, we identified key genes that were strongly linked to seed nutrient composition, providing insights into the potential molecular determinants influencing the observed variations in seed quality among the cultivars under investigation.

The findings revealed that in the D cultivar, compared with the common group, 35 DEGs were strongly associated with such seed quality traits. Specifically, the expression patterns of these 35 DEGs were positively correlated with increased levels of Ca, fat, Fe, protein, and soluble sugars in the seeds. Conversely, their expression was negatively correlated with higher

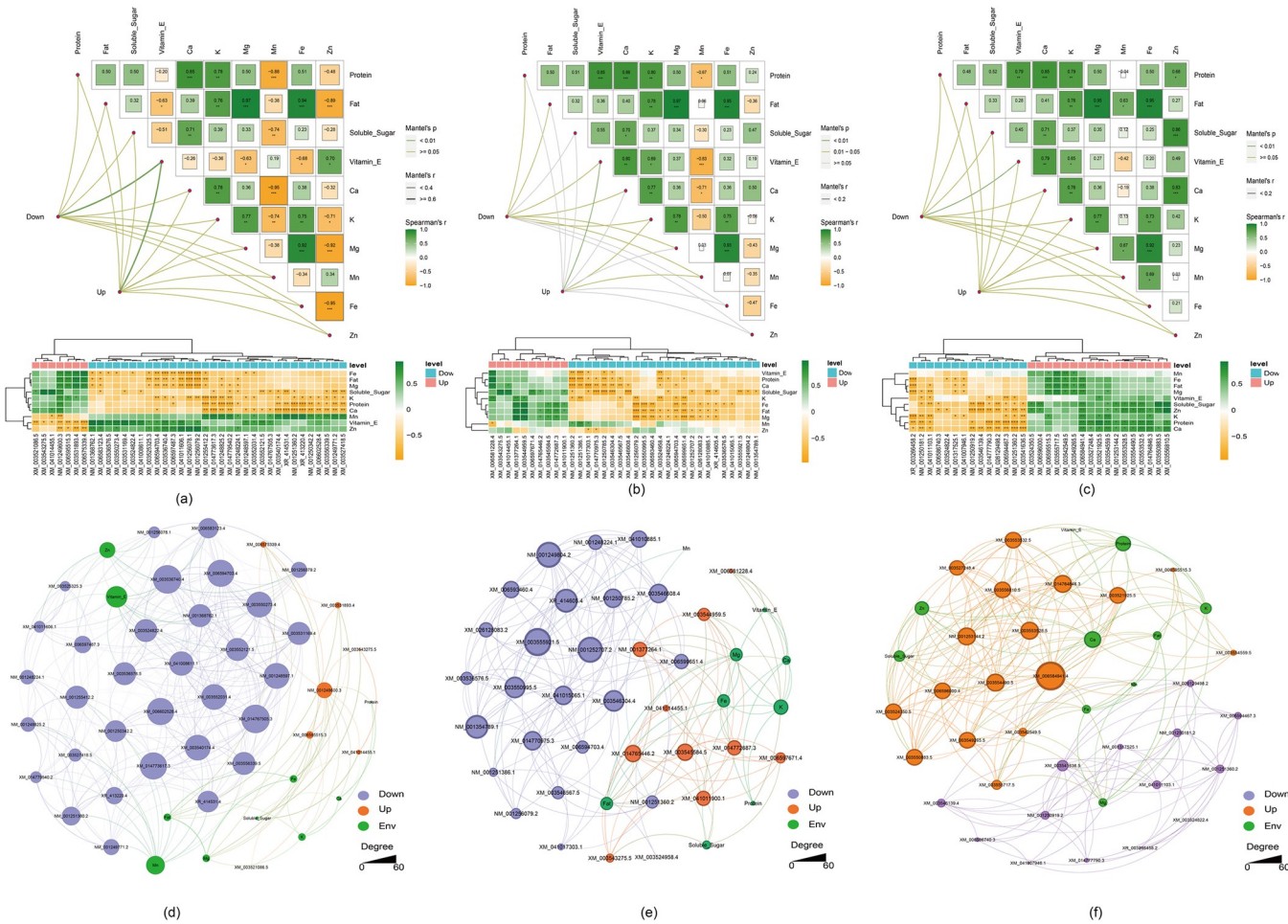

**Fig 5. Association analysis between key DEGs and nutritional profiles.** (**a-c**) Correlations between quality traits and DEGs in three fresh maodou varieties. The pairwise comparisons of quality indices are visually represented, utilizing a color gradient to illustrate Spearman's correlation coefficients. The associations between DEGs and each trait were determined via partial Mantel tests. The curve width is proportional to Mantel's r statistic for the corresponding distance correlations, and the curve color (see the legend on the right) denotes the statistical significance based on Spearman's p value. Correlation analysis between the quality traits and DEGs. Yellow boxes represent negative correlations, whereas green boxes represent positive correlations (Spearman's correlation). The black asterisks indicate statistical significance: *, p value < 0.05; **, p value <0.01; ***, p value <0.001. (**d-f**) A co-occurrence network displays the correlations between quality traits and the DEGs of three fresh maodou. The size of each node is proportional to the number of connections. The nodes were colored according to upregulated, downregulated and quality traits, which were determined via Gephi.

concentrations of K, Mg, Mn, vitamin E, and Zn (**Fig 6A**). When the E cultivar was compared with the common cultivar, our analysis revealed 27 DEGs strongly correlated with the seed quality traits. The expression of these 27 DEGs was positively associated with higher levels of Ca, fat, Fe, Mn, protein, and soluble sugars but negatively correlated with K, vitamin E, and Zn concentrations (**Fig 6B**). Compared with the common cultivar, 75 DEGs were positively associated with higher levels of calcium, fat, iron, protein, and soluble sugars in the seeds of the F cultivar. However, their expression did not significantly correlate with the concentrations of K, Mg, Mn, vitamin E, or Zn (**Fig 6C**).

### 3.6 Transcriptome validation of DEGs via qRT–PCR

The clustering heatmap analysis of gene expression profiles (**Fig 7**) across four distinct soybean cultivars, comprising three common cultivars (HN551, HN562, and HN63) and the fresh

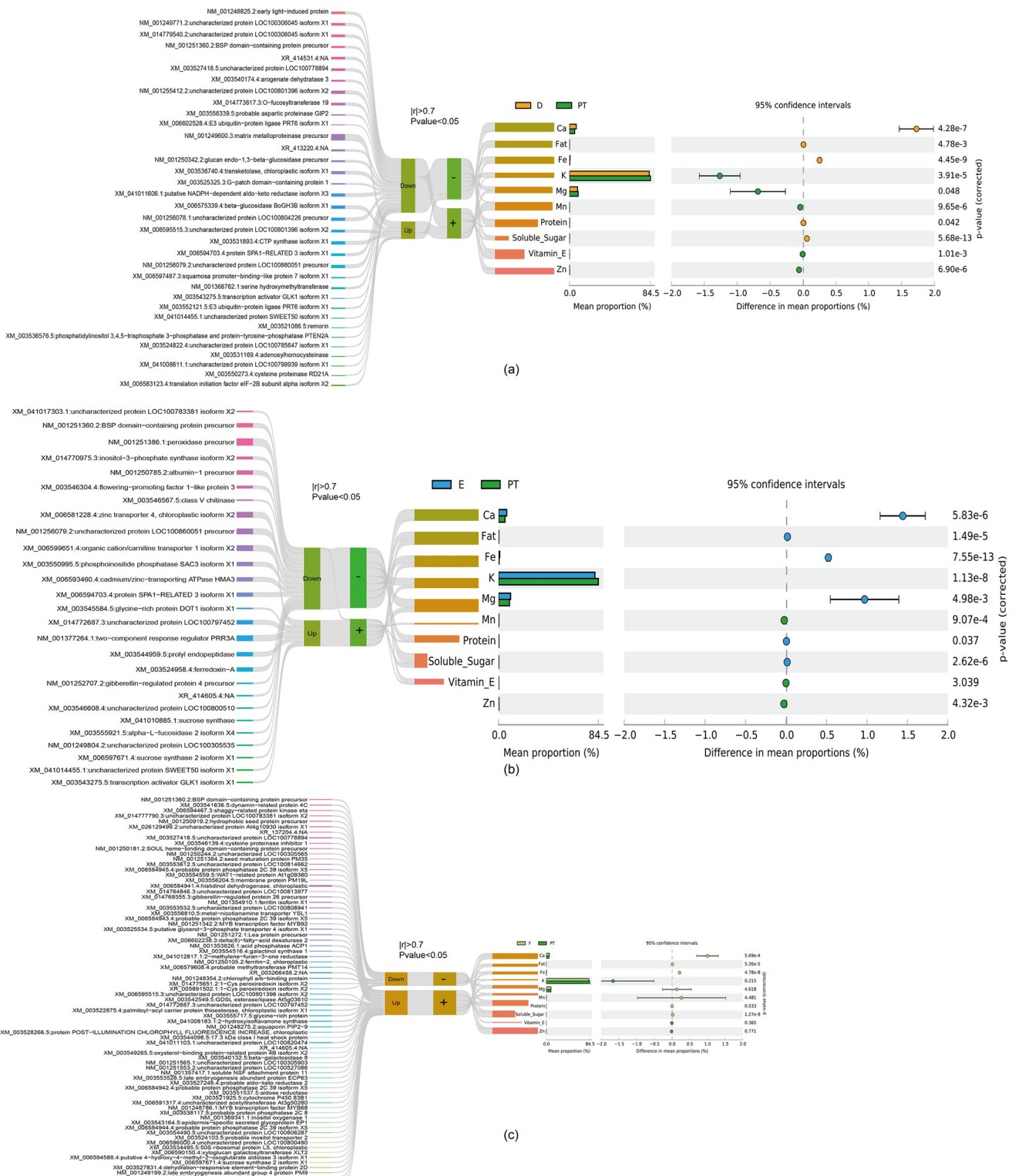

**Fig 6.** Sankey diagrams illustrating DEGs and their associations with seed quality traits in the three fresh maodou soybean cultivars (**a**: cultivar D, **b**: cultivar E, **c**: cultivar F) compared with the common group. The diagrams comprise five columns, from left to right: individual DEGs, clusters of co-expressed DEGs, correlations meeting the criteria of Spearman's p value < 0.05 and |rho| > 0.7, and the specific seed quality parameters exhibiting significant differences compared with those of the common cultivars. The flow of the Sankey diagrams represents the relationships between the identified DEGs, their expression patterns within distinct clusters, and their potential contributions to the variations observed in seed nutrient composition, such as protein, fat, soluble sugar, vitamin, and mineral levels, compared with their common counterparts.

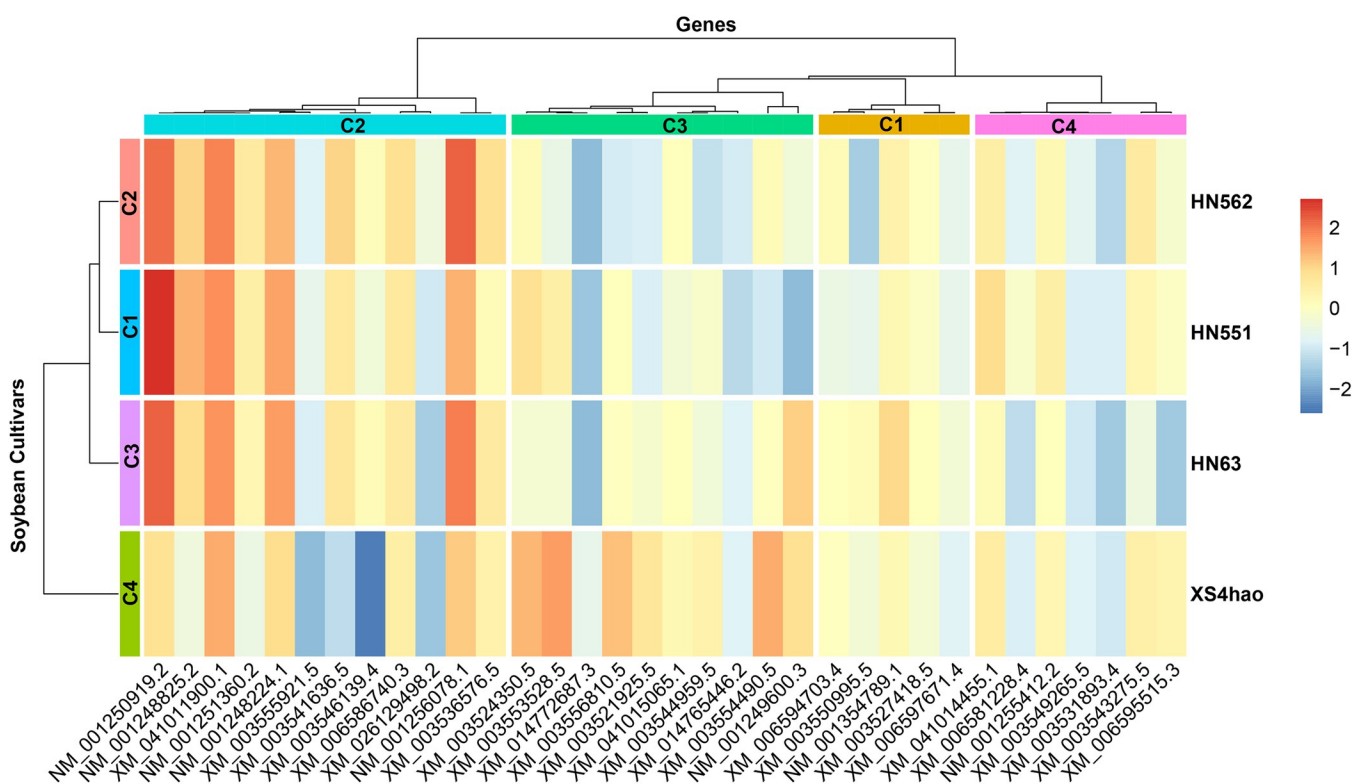

**Fig 7. Hierarchical clustering and heatmap visualization of gene expression profiles in four vegetable soybean cultivars.** The heatmap displays the expression patterns of 34 genes (columns) across the cultivars HN551, HN562, HN63, and XS4hao (rows). The expression levels are represented by a color gradient, with blue indicating low expression and red indicating high expression. Both genes and cultivars were subjected to hierarchical clustering on the basis of their expression profiles, as depicted by the dendrogram. Four distinct gene clusters (C1-C4) and cultivar clusters were identified, highlighting co-expressed gene sets and cultivar groups with similar transcriptomic signatures.

maodou cultivar (XS4hao), revealed intricate patterns that provide valuable insights into the molecular underpinnings of cultivar-specific traits. The hierarchical clustering approach employed in this study enabled the identification of distinct subgroups among both cultivars and the 34 genes under investigation, highlighting the complex relationships and variations in gene expression patterns.

Heatmap analysis revealed distinct patterns of gene expression among four soybean cultivars: XS4hao, HN63, HN551, and HN562. Hierarchical clustering was performed on both the cultivars and the 34 genes, resulting in clear subgroupings that highlighted the relationships and differences in gene expression profiles across these cultivars. The hierarchical clustering of cultivars revealed four distinct clusters: C1, C2, C3, and C4. Heatmap clustering revealed that the cultivars HN551 (C1), HN562 (C2), and HN63 (C3) were clustered together apart from the cultivar XS4hao. These clusters suggest significant variability in gene expression profiles among the cultivars, reflecting their genetic and physiological diversity. The gene expression profiles were also grouped into four distinct clusters: C1, C2, C3, and C4. The genes in Cluster C1 included XM_006597671.4, XM_003527418.5, NM_001354789.5, XM_003550995.5, and XM_006594703.4, whereas those in Cluster C2 included XM_003536576.5, NM_001256078.1, XM_026129498.2, XM_006586740.3, XM_003546139.4, XM_003541636.5, XM_003555921.5, NM_001248224.1, NM_001251360.2, XM_041011900.1, NM_001248825.2, and NM_001250919.2. The genes in cluster C3 included NM_001249600.3, XM_003554490.5, XM_014765446.2, XM_003544959.5, XM_041015065.1, XM_003521925.5, XM_003556810.5,

XM_014772687.3 XM_003553528.5, and XM_003512350.5, whereas those in cluster C4 included XM_006595515.3, XM_003543275.5, XM_003531893.4, XM_003549265.5, NM_001255412.2, XM_006581228.4, and XM_041014455.1.

The genes in cluster C2 presented high expression levels with HN551, HN562, and HN63 but mixed variable expression levels with the cultivar XS4hao. The Cluster C3 genes presented relatively low expression levels across most cultivars, except for some genes whose expression was relatively high in XS4hao. The Cluster C1 and C4 genes presented moderate expression levels across all cultivars. These gene clusters indicate specific sets of genes whose expression is upregulated or downregulated in particular cultivars, suggesting potential targets for further functional analysis. Certain genes, such as NM_001250919.2. showed very high expression levels in specific cultivars, such as HN551, suggesting their potential roles in the unique traits of this cultivar. Conversely, genes such as XM_003546139.4 presented lower expression levels in cultivars such as XS4hao, indicating different regulatory controls or functional roles in these cultivars.

## 4. Discussion

This study investigated the nutrient contents of seeds from six types of vegetable soybean (maodou), commonly known as fresh or common soybeans. These cultivars demonstrated significant variation in their concentrations of protein, fat, soluble sugars, vitamin E, calcium, potassium, magnesium, manganese, iron, and zinc, reflecting their nutritional diversity. Such diversity is linked to observed differences in plant architecture, growth, development, and chemical composition [40]. Previous research has also highlighted the richness of maodou soybeans in essential nutrients vital for human health [10,13,20]. The differences in protein and fat, vitamin E, magnesium, and iron contents across the cultivars highlight the significance of choosing cultivars based on their nutritional qualities to meet specific application needs. Additionally, the observed variations in the contents of minerals, including Mg, Mn, Fe, and Zn, further highlight the nutritional diversity among the soybean cultivars studied. These findings underscore the importance of understanding and utilizing the distinct nutrient profiles of vegetable soybean cultivars to develop targeted breeding programs that enhance nutritional traits for specific dietary or industrial purposes [41,42].

Previous investigations on vegetable soybean have revealed significant nutritional characteristics and applications. Studies in Dharwad, India demonstrated that lines AGS406 and AGS447 exhibited enhanced sweetness compared to the grain soybean variety JS-335 in organoleptic assessments [43]. Further research by Maruthi et al. [44] identified elevated concentrations of crucial amino acids (glutamic acid, alanine, histidine, and asparagine) that contribute to the distinctive flavor and sweetness of vegetable soybeans. The presence of isoflavones, essential nutrients in vegetable soybeans, has been documented in various forms including glycosides: genistin (β-glucosides), daidzin (acetyl-β-glucosides), glycitin (malonyl-β-glucosides), and aglucones. The applications of vegetable soybean extend beyond human consumption. In poultry production, feed supplemented with vegetable soybean isoflavones (3 g/kg of feed) has shown beneficial effects on grill chicken carcass quality, specifically reducing fat percentage and cholesterol levels [45]. Additionally, industrial byproducts such as unmarketable pods and beans have been repurposed through fermentation to produce vegetable soybean chips for poultry feed. The fermented vegetable soybean meal has demonstrated positive effects on black-boned chickens, enhancing both immunity and muscle mass. In food processing applications, fermented soymilk from vegetable soybeans has proven to be an effective matrix for probiotic bacteria delivery while offering reduced levels of non-digestible oligosaccharides [46]. Notably, second-generation soy products contain minimal bioactive components, while

traditional soy preparations maintain moderate levels, resulting in limited health benefits with negligible risk of adverse effects [47].

Compared with mature soybeans, vegetable soybeans contain greater amounts of important nutrients such as iron, protein, folate, zinc, linoleic acid, and linolenic acid [48–50]. This makes them useful for making diets more varied and fixing nutritional deficiencies [9,17,18]. The incorporation of maodou into diets can not only provide essential nutrients but also support biofortification efforts, assist in the development of sustainable diets, and contribute to the fight against malnutrition, highlighting their importance in promoting health and well-being [50].

The comprehensive analysis of transcriptomic profiles through RNA-Seq technology provides valuable insights into the genetic mechanisms governing phenotypic variations among different soybean cultivars [2,51]. Through the examination of gene expression patterns, we aimed to elucidate the molecular underpinnings contributing to the observed phenotypic diversity. The comparative transcriptome analysis conducted in this study effectively identified DEGs between fresh and common soybean cultivars, thereby providing valuable insights into the molecular mechanisms underlying their phenotypic differences. The E and F cultivars presented the most significant DEGs, with 2,304 and 1,145 DEGs, respectively, suggesting more pronounced genetic differentiation between these fresh maodou cultivars and the common cultivars. The magnitude of gene expression differences varied across various comparison groups, as evidenced by further investigation of the main DEGs. The phenotypic differences between fresh and common cultivars are likely to be significantly influenced by these main genes, and further investigations are warranted [52]. Moreover, the construction of Venn diagrams and heatmaps revealed overlapping DEGs among the three comparison groups, indicating a core set of genes that may be responsible for the distinct characteristics of fresh maodou cultivars. The identified DEGs and core sets of genes can be further investigated to elucidate their functional roles in soybean growth, development, and nutritional composition, providing valuable resources for breeding programs aimed at developing cultivars with enhanced nutritional profiles tailored to specific dietary needs or industrial applications.

Thus, functional enrichment analysis of DEGs in fresh maodou cultivars using the GO and KEGG databases has offered valuable insights into the molecular mechanisms driving their phenotypic differences compared to common cultivars. GO enrichment analysis revealed that the DEGs in the D and E cultivars were significantly enriched in several GO terms underpinning a potential role in the regulation of metabolic processes. For the F cultivar, the upregulated DEGs were significantly enriched in GO terms related to response to desiccation, maintenance of seed dormancy, and peroxidase activity, suggesting a potential role in the regulation of seed dormancy and stress responses. Importantly, KEGG enrichment analysis revealed that in sample E, compared with the common, 166 DEGs were significantly enriched in ko00940 (phenylpropanoid biosynthesis), whereas in sample F, compared with the common, 154 DEGs were significantly enriched across six KEGG pathways. These results indicate that these pathways may be involved in the regulation of metabolic processes and stress responses in fresh maodou cultivars. These results can be further investigated to clarify the functional functions of these genes and pathways in the nutritional composition, growth, and development of soybean.

Interestingly, studies examining how DEGs are linked to the nutritional profiles of different maodou cultivars have shown that there is a strong link between differences in gene expression patterns and differences in the nutritional profiles of these different cultivars [53]. Both the upregulated and downregulated DEGs were significantly correlated with various nutritional components, including protein, fat, soluble sugars, vitamin E, and mineral elements such as calcium, potassium, magnesium, manganese, iron, and zinc. For example, in cultivar D, the set of 32 downregulated DEGs had strong negative relationships with iron, manganese, vitamin E,

soluble sugars, calcium, potassium, magnesium, and iron. On the other hand, these genes are strongly positively related to manganese, vitamin E, and zinc. These findings suggest that downregulating these genes may increase plant vitamin E and that upregulating them may increase other nutrients. The key downregulated DEGs found in cultivar D suggest that the differential expression of these genes may affect the accumulation of various nutrients, such as minerals and vitamin E. For cultivar F, the identified set of upregulated DEGs associated with specific nutrients suggests that the upregulation of these genes may contribute to enhanced nutritional quality, particularly in terms of protein, mineral, and vitamin E contents. Furthermore, the co-occurrence network constructed via Spearman correlation analysis provides insights into the potential reciprocal interactions between those key DEGs and the quality parameters studied. The nutritional quality of vegetative soybean can be enhanced through genetic manipulation or further investigation of these genes or gene networks [54].

In this study, we conducted a comprehensive analysis of the nutritional profiles of fresh and common soybean cultivars, revealing significant differences in their metabolic pathways that contribute to variations in seed quality. Key metabolic pathways such as lipid metabolism, amino acid biosynthesis, and carbohydrate metabolism were explored to understand the underlying molecular mechanisms. For instance, the expression of genes involved in fatty acid biosynthesis was notably higher in fresh cultivars, suggesting increased fat accumulation, while specific DEGs related to protein synthesis indicated enhanced protein content in these cultivars. Additionally, we observed that genes associated with sugar metabolism were differentially expressed, which may explain the elevated levels of soluble sugars in fresh soybeans. Functional enrichment analyses highlighted the overrepresentation of pathways related to lipid and protein metabolism in fresh cultivars, further emphasizing the biochemical shifts that occur during seed development. By integrating these findings, our study provides valuable insights into the metabolic changes between fresh and common soybean cultivars, offering a molecular basis for understanding their distinct nutritional profiles.

The gene expression profiles across maodou cultivars revealed intricate patterns that provide valuable insights into the molecular underpinnings of their unique nutritional profiles [41]. The hierarchical clustering approach revealed a close relationship between HN551, HN562, and HN63 gene expression patterns, forming a distinct group separate from XS4hao. This segregation suggests significant variability in the genetic and physiological characteristics governing nutrient accumulation and metabolism between the common cultivars and the fresh maodou cultivars [41,55]. Interestingly, the clustering of genes into distinct groups revealed co-expressed gene sets that may be directly involved in the biosynthesis, uptake, translocation, and accumulation of specific nutrients [41,55]. For example, the gene cluster exhibiting relatively high expression levels in HN551 could be associated with pathways involved in Mn uptake, translocation, and storage, potentially contributing to the elevated Mn content observed in this cultivar. Similarly, the gene cluster upregulation in HN562 and HN63 could be linked to mechanisms governing Zn and vitamin E metabolism, respectively, leading to their increased accumulation. Genes displaying remarkably high or low expression levels in XS4hao could be implicated in shaping its elevated levels of protein, fat, vitamin E, Mg, and Fe. Unraveling these cultivar-specific gene expression signatures and their associated regulatory mechanisms could aid in elucidating the underlying genetic determinants of desirable nutritional traits and informing targeted breeding strategies for enhancing the nutritional quality of soybean cultivars [13].

## 5. Conclusion

This study provides a comprehensive analysis of nutritional profiles and transcriptomic landscapes across six vegetable soybean cultivars, revealing significant variations in essential

nutrient concentrations. Comparative transcriptome analysis identified numerous differentially expressed genes (DEGs) between fresh and common cultivars, with enrichment in crucial biological processes related to metabolic regulation and seed development. Strong correlations between DEG expression patterns and nutrient profiles suggest these genes play pivotal roles in regulating nutrient accumulation and metabolism. Hierarchical clustering of gene expression profiles revealed distinct patterns between common and fresh maodou cultivars, underscoring the genetic and physiological variability governing nutrient accumulation. These findings bridge the gap between phenotypic observations of nutritional diversity and underlying genetic mechanisms in vegetable soybean cultivars. The identified DEGs and their associations with specific nutrient profiles offer promising targets for marker-assisted selection and genetic engineering approaches aimed at enhancing soybean nutritional quality. Future research should focus on validating these findings through metabolomic analyses and functional genomics studies. Additionally, obtaining and analyzing complete transgenic plants overexpressing or silencing key identified DEGs will be crucial for conclusively determining their roles in nutrient accumulation and metabolism. These further investigations will provide more solid evidence to support our conclusions and accelerate the development of nutritionally enhanced soybean cultivars.

## Supporting information

**S1 Table. RT-PCR primers sequence.**
(XLSX)

**S2 Table. RNA-sequencing libraries.**
(XLSX)

**S3 Table. RNA libraries Quality control measures, including filtration.**
(XLSX)

**S4 Table. Subsequent alignment of the filtered reads to the soybean reference genome.**
(XLSX)

## Author Contributions

**Conceptualization:** Chunlei Zhang, Min Zhang, Bixian Zhang, Wencheng Lu.

**Data curation:** Rongqiang Yuan, Kezhen Zhao, Sobhi F. Lamlom.

**Formal analysis:** Chunlei Zhang, Rongqiang Yuan, Kezhen Zhao, Min Zhang, Ahmed M. Abdelghany, Sobhi F. Lamlom, Honglei Ren.

**Funding acquisition:** Rongqiang Yuan, Min Zhang, Sobhi F. Lamlom, Bixian Zhang.

**Investigation:** Xueyang Wang, Rongqiang Yuan, Xiulin Liu, Kezhen Zhao, Min Zhang.

**Methodology:** Chunlei Zhang, Rongqiang Yuan, Xiulin Liu.

**Project administration:** Xueyang Wang, Xiulin Liu, Bixian Zhang, Jia Liu, Wencheng Lu.

**Resources:** Xiulin Liu, Bixian Zhang, Qiang Qiu, Jia Liu, Wencheng Lu, Honglei Ren.

**Software:** Chunlei Zhang, Xiulin Liu, Fengyi Zhang, Kezhen Zhao, Min Zhang, Ahmed M. Abdelghany, Sobhi F. Lamlom, Qiang Qiu, Jia Liu, Honglei Ren.

**Supervision:** Fengyi Zhang, Min Zhang, Ahmed M. Abdelghany, Bixian Zhang, Qiang Qiu, Jia Liu, Wencheng Lu, Honglei Ren.

**Validation:** Xueyang Wang, Fengyi Zhang, Ahmed M. Abdelghany, Bixian Zhang, Qiang Qiu, Jia Liu, Honglei Ren.

**Visualization:** Fengyi Zhang, Kezhen Zhao, Ahmed M. Abdelghany, Bixian Zhang, Jia Liu, Honglei Ren.

**Writing – original draft:** Xueyang Wang, Chunlei Zhang, Fengyi Zhang, Kezhen Zhao, Ahmed M. Abdelghany, Sobhi F. Lamlom, Honglei Ren.

**Writing – review & editing:** Fengyi Zhang, Ahmed M. Abdelghany, Sobhi F. Lamlom, Bixian Zhang, Wencheng Lu, Honglei Ren.

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
