## [Decision Letter · Decision Letter 0]

21 Oct 2024

PONE-D-24-40891Transcriptome Profiling Uncovers Differentially Expressed Genes Linked to Nutritional Quality in Vegetable SoybeanPLOS ONE

Dear Dr. Ren,

Thank you for submitting your manuscript to PLOS ONE. After careful consideration, we feel that it has merit but does not fully meet PLOS ONE’s publication criteria as it currently stands. Therefore, we invite you to submit a revised version of the manuscript that addresses the points raised during the review process.

We look forward to receiving your revised manuscript.

Kind regards,

Aminallah Tahmasebi

Academic Editor

PLOS ONE

Journal requirements: When submitting your revision, we need you to address these additional requirements. 1. Please ensure that your manuscript meets PLOS ONE's style requirements, including those for file naming. The PLOS ONE style templates can be found at https://journals.plos.org/plosone/s/file?id=wjVg/PLOSOne_formatting_sample_main_body.pdf and https://journals.plos.org/plosone/s/file?id=ba62/PLOSOne_formatting_sample_title_authors_affiliations.pdf 2. Please include a caption for figure 2. 3. We note that the grant information you provided in the ‘Funding Information’ and ‘Financial Disclosure’ sections do not match.  When you resubmit, please ensure that you provide the correct grant numbers for the awards you received for your study in the ‘Funding Information’ section. 4. Thank you for stating the following financial disclosure:  [funded by Scientific Research Business Expenses of Heilongjiang Scientific Research Institutes (Grant No. CZKYF2021-2-C030; Grant No. CZKYF2024-1-A003; Grant No. CZKYF2022-1-A002; Grant No. CZKYF2023-1-A001); Innovation Leaping Project in Heilongjiang Academy of Agricultural Science (Grant No. CX23ZD04; Grant No. CX23TS04; Heilongjiang Natural Science Foundation joint guidance project（LH2020C093）.].  Please state what role the funders took in the study.  If the funders had no role, please state: ""The funders had no role in study design, data collection and analysis, decision to publish, or preparation of the manuscript."" If this statement is not correct you must amend it as needed. Please include this amended Role of Funder statement in your cover letter; we will change the online submission form on your behalf. 5. Thank you for stating the following in the Acknowledgments Section of your manuscript: [This study was funded by the Scientific Research Business Expenses of Heilongjiang Scientific Research Institutes (Grant No. CZKYF2021-2-C030; Grant No. CZKYF2024-1-A003; Grant No. CZKYF2022-1-A002; Grant No. CZKYF2023-1-A001); Innovation Leaping Project in Heilongjiang Academy of Agricultural Science (Grant No. CX23ZD04; Grant No. CX23TS04; Heilongjiang Natural Science Foundation joint guidance project（LH2020C093)]We note that you have provided funding information that is not currently declared in your Funding Statement. However, funding information should not appear in the Acknowledgments section or other areas of your manuscript. We will only publish funding information present in the Funding Statement section of the online submission form. Please remove any funding-related text from the manuscript and let us know how you would like to update your Funding Statement. Currently, your Funding Statement reads as follows:  [funded by Scientific Research Business Expenses of Heilongjiang Scientific Research Institutes (Grant No. CZKYF2021-2-C030; Grant No. CZKYF2024-1-A003; Grant No. CZKYF2022-1-A002; Grant No. CZKYF2023-1-A001); Innovation Leaping Project in Heilongjiang Academy of Agricultural Science (Grant No. CX23ZD04; Grant No. CX23TS04; Heilongjiang Natural Science Foundation joint guidance project（LH2020C093）.].   Please include your amended statements within your cover letter; we will change the online submission form on your behalf. 6. Please upload a copy of table S2 and table S3 to which you refer in your text on page 8. Please amend the file type to 'Supporting Information'. If the Supplementary file is no longer to be included as part of the submission please remove all reference to it within the text. 7. Please include captions for your Supporting Information files at the end of your manuscript, and update any in-text citations to match accordingly. Please see our Supporting Information guidelines for more information: http://journals.plos.org/plosone/s/supporting-information. 

Additional Editor Comments:

Dear Dr. Ren,

Thank you for submitting your manuscript to PLOS One.

I have completed my evaluation of your manuscript. The reviewers recommend reconsideration of your manuscript following revision and modification. I invite you to resubmit your manuscript after addressing the comments below. Please resubmit your revised manuscript by October 29, 2024.

When revising your manuscript, please consider all issues mentioned in the reviewers' comments carefully: please outline every change made in response to their comments and provide suitable rebuttals for any comments not addressed. Please note that your revised submission may need to be re-reviewed.

Reviewers' comments:

Reviewer's Responses to Questions

**Comments to the Author**

1. Is the manuscript technically sound, and do the data support the conclusions?

Reviewer #1: No

Reviewer #2: Yes

2. Has the statistical analysis been performed appropriately and rigorously? 

Reviewer #1: No

Reviewer #2: Yes

3. Have the authors made all data underlying the findings in their manuscript fully available?

Reviewer #1: No

Reviewer #2: Yes

4. Is the manuscript presented in an intelligible fashion and written in standard English?

Reviewer #1: No

Reviewer #2: Yes

5. Review Comments to the Author

Reviewer #1: The number of repetitions for the sample in the field experiment is insufficient. In a controlled greenhouse setting, three repetitions may be adequate for a simple test. However, in the field, where numerous uncontrolled factors can influence gene expression, using only three repetitions for each cultivar is not acceptable.

Reviewer #2: I recommend this manuscript for publication in PLOS ONE following minor revisions. The revisions are mostly aimed at interpretation of the results and clarity of the language such that it would be accessible to a larger audience, hence a greater contribution.

• The authors have identified nutritional traits-related DEGs effectively, but such understanding could be even further enriched when a more mechanistic explanation is provided about how DEGs regulate nutrient composition. For example, there is a place for further strengthening the discussion in lines 236-245 by complementing the information with more biological context concerning the functional implication of such genes.

• Similarly, the pathways and genes identified from KEGG enrichment analyses in lines 270-278 could be elaborated to further account for the biological importance of DEGs with respect to nutrient biosynthesis and the stress response.

• Although the manuscript contains lots of data on nutritional composition, no detailed comparison of metabolic pathways between fresh and common soybean cultivars is done in the manuscript. On lines 290-299, to observe which metabolic changes have taken place, more contexts need to be discussed.

• The manuscript does comply with the PLOS ONE data-sharing policy. Lines 126-128 would also read better if the accession numbers or links for RNA-seq were placed directly within the text to make the data more readable to interested readers and researchers who might want to scrutinize it.

• The manuscript is generally well-written, although some places require minor refinement with regard to chosen language in order to enhance clarity. For example, complicated sentences in the introduction should be simplified for easier reading (lines 34-40).

• The manuscript is well-structured and well-organized with the discussion, particularly lines 384-394, showing a need for comparison to related studies on vegetable soybean varieties. More context background would be able to emphasize the relevance and novelty of the study by showing how these findings relate to existing literature.

• The study will interest agricultural researchers and breeders alike, especially with the current upsurge in functional foods. The authors could do a better explanation on practical applications and limitations of this research into breeding programs on lines 431-435 to place more emphasis on the importance of the study in fronting nutritionally improved soybean varieties.

6. PLOS authors have the option to publish the peer review history of their article (what does this mean?). If published, this will include your full peer review and any attached files.

Reviewer #1: No

Reviewer #2: **Yes: **Pranavkumar Gajjar

---

## [Author Response · Author response to Decision Letter 0]

25 Oct 2024

Reviewer 1

The number of repetitions for the sample in the field experiment is insufficient. In a controlled greenhouse setting, three repetitions may be adequate for a simple test. However, in the field, where numerous uncontrolled factors can influence gene expression, using only three repetitions for each cultivar is not acceptable.

Response: Thank you for your insightful comments regarding the number of repetitions used in the field experiment. We sincerely apologize for not providing a detailed explanation of our sampling methodology in the initial version of the manuscript, which may have led to concerns about the experimental design. We would like to clarify our comprehensive sampling strategy, which was more robust than initially described. Our experiment was established using a randomized complete block design with three blocks, where each cultivar was planted in four-row plots to account for field heterogeneity. For each block, we collected three independent samples which were then pooled to create one representative biological replicate. This means that while we ultimately worked with three biological replicates per cultivar, each replicate actually represents a composite sample derived from three distinct sampling points within each block. Therefore, our sampling effectively covered nine distinct sampling points per cultivar across the field. This composite sampling approach was specifically designed to capture and account for field variation while maintaining statistical rigor. The pooling strategy helps minimize micro-environmental effects within plots while still maintaining the statistical power of three true biological replicates. We believe this sampling methodology provides a more representative assessment of gene expression patterns than simple three-point sampling, as each biological replicate integrates multiple sampling points from within each block. This approach, combined with our randomized complete block design, effectively addresses the concerns about field variation while maintaining the statistical validity necessary for transcriptome analysis. We also update that in the manuscript in lines 159:164

Reviewer 2

I recommend this manuscript for publication in PLOS ONE following minor revisions. The revisions are mostly aimed at interpretation of the results and clarity of the language such that it would be accessible to a larger audience, hence a greater contribution.

Response: Dear Reviewer, thank you for your positive recommendation of our manuscript for publication in PLOS ONE, pending minor revisions. We greatly appreciate your feedback regarding the interpretation of our results and the clarity of our language.

• The authors have identified nutritional traits-related DEGs effectively, but such understanding could be even further enriched when a more mechanistic explanation is provided about how DEGs regulate nutrient composition. For example, there is a place for further strengthening the discussion in lines 236-245 by complementing the information with more biological context concerning the functional implication of such genes.

Response: Thank you for your constructive feedback regarding our discussion of DEGs related to nutritional traits. We appreciate your suggestion to provide a more mechanistic explanation of how these DEGs regulate nutrient composition. We updated the text highlighted with complementing the information with more biological context concerning the functional implication of such genes (Lines 241:285)

• Similarly, the pathways and genes identified from KEGG enrichment analyses in lines 270-278 could be elaborated to further account for the biological importance of DEGs with respect to nutrient biosynthesis and the stress response.

Response: Thank you for your insightful comments regarding the KEGG enrichment analyses presented in our manuscript. In response to your feedback, we have expanded the discussion in lines 310-316 to provide a more comprehensive overview of the identified pathways and their specific roles in regulating nutrient composition and stress resilience in vegetable soybean cultivars (Lines 307:316).

• Although the manuscript contains lots of data on nutritional composition, no detailed comparison of metabolic pathways between fresh and common soybean cultivars is done in the manuscript. On lines 290-299, to observe which metabolic changes have taken place, more contexts need to be discussed.

Response: In response to your suggestion, we have expanded the discussion to include a comprehensive comparison of the metabolic pathways associated with both fresh and common soybean cultivars (Lines 467:477).

• The manuscript does comply with the PLOS ONE data-sharing policy. Lines 126-128 would also read better if the accession numbers or links for RNA-seq were placed directly within the text to make the data more readable to interested readers and researchers who might want to scrutinize it.

Response: we have revised lines 126-128 to incorporate the relevant links directly within the text.

• The manuscript is generally well-written, although some places require minor refinement with regard to chosen language in order to enhance clarity. For example, complicated sentences in the introduction should be simplified for easier reading (lines 34-40).

Response: Thank you for your positive feedback regarding the overall quality of our manuscript and for highlighting areas that require minor refinements. In response to your comment, we have reviewed lines 34-40 and simplify any complicated sentences to improve readability (Lines 33:37).

• The manuscript is well-structured and well-organized with the discussion, particularly lines 384-394, showing a need for comparison to related studies on vegetable soybean varieties. More context background would be able to emphasize the relevance and novelty of the study by showing how these findings relate to existing literature.

Response: we expanded this section by integrating comparisons to relevant literature that explores similar aspects of vegetable soybean research. We will highlight key findings from these studies, drawing parallels and contrasts with our results (Lines 402:417)

• The study will interest agricultural researchers and breeders alike, especially with the current 

upsurge in functional foods. The authors could do a better explanation on practical applications 

and limitations of this research into breeding programs on lines 431-435 to place more emphasis on the importance of the study in fronting nutritionally improved soybean varieties. 

Response: We appreciate your recognition of the study's relevance to agricultural researchers and breeders, especially in the context of the growing interest in functional foods. We provided more detailed explanation of the practical applications of our findings in the Conclusion section (Lines 501-508).

---

## [Editor Report · Decision Letter 1]

29 Oct 2024

Transcriptome Profiling Uncovers Differentially Expressed Genes Linked to Nutritional Quality in Vegetable Soybean

PONE-D-24-40891R1

Dear Dr. Ren,

We’re pleased to inform you that your manuscript has been judged scientifically suitable for publication and will be formally accepted for publication once it meets all outstanding technical requirements.

Kind regards,

Aminallah Tahmasebi

Academic Editor

PLOS ONE

Additional Editor Comments (optional):

The manuscript is suitable for publication in its present form.
---

## [Editor Report · Acceptance letter]

4 Nov 2024

PONE-D-24-40891R1 

PLOS ONE

Dear Dr. Ren, 

I'm pleased to inform you that your manuscript has been deemed suitable for publication in PLOS ONE. Congratulations! Your manuscript is now being handed over to our production team.

Kind regards, 

on behalf of

Dr. Aminallah Tahmasebi 

Academic Editor

PLOS ONE